# A Coevolution Model of the Coupled Society—Water Resources—Environment Systems: An Application in a Case Study in the Yangtze River Economic Belt, China

Haoyuan Liu [1,2], Xiang Zhang [1,2,*], Shiyong Tao [1,2], Xi Xiao [1,2], Keyi Wu [1,2] and Jun Xia [1,2]

1   State Key Laboratory of Water Resources and Hydropower Engineering Science, Wuhan University, Wuhan 430072, China
2   Hubei Key Laboratory of Water System Science for Sponge City Construction, Wuhan University, Wuhan 430072, China
*   Correspondence: zhangxiang@whu.edu.cn

**Abstract:** Interactions among society, water resources, and environment systems have become increasingly prominent with the progressively far-reaching impact of human activities. Therefore, this paper aims to construct a co-evolution model to establish the mutual feedback relationship among society, water resources, and environment from the perspective of socio-hydrology. Firstly, social factors such as environmental sensitivity, environmental protection awareness, and technological level are introduced to this model to describe the coevolutionary trajectory of society, water resources and environment subsystems. Then, this model is implemented in 11 provincial administrative regions in the Yangtze River Economic Belt, and the degree of coordination of their coupling is evaluated. Results show that the water-use efficiency of each provincial administrative region in the Yangtze River Economic Belt gradually increases during the forecast period. The coupling-coordinated degree of each provincial administrative region of the Yangtze River Economic Belt has greatly improved during the 14th Five-Year Plan period, reflecting that policy support has played a significant role in the coordinated development of the Yangtze River Economic Belt. The dynamic fluctuation process of environmental sensitivity effectively depicts the co-evolution process of the coupling system, which provides a reference for the subsequent exploration and cognition of the human-water coevolutionary mechanism.

**Keywords:** co-evolution; socio-hydrology; environmental sensitivity; environmental protection awareness; coupling coordination degree; Yangtze River Economic Belt





## 1. Introduction

During the Anthropocene, human activities have exerted an unprecedented impact on the natural environment [1]. Freshwater is one of the most critical and valuable natural resources for maintaining a virtuous cycle of social production and environmental evolution [2]. However, since the industrial revolution, explosive population growth and high-intensity economic activity have caused and accelerated the exploitation of water resources, resulting in and exacerbating water contamination [3].Water problems such as water efficiency [4], drinking water safety [5], and sewage discharge [6] brought about by the development of human society have become increasingly conspicuous and interconnected. It is now widely recognized and accepted that society, water resources and the environment interact together and are studied as a whole (called Society Water and Environment (SWE) coupling system in this research) [7]. With the increasing intensity of disturbances to the natural environment by human activities, the vincula among water resources, economy, society, institution, environment and other elements in the human-nature coupling system are getting closer and closer [8]. Global environmental issues are becoming increasingly prominent under the stress of intense water-related human activities, and

addressing this issue has become a primary concern for water resources management [9]. If effective management measures are not implemented, the competition between social development, water use and environmental protection will intensify in the future. How to consider these vincula and competitions for sustainable human societies has become a common concern for decision makers and water scientists [10].

In order to reveal the complicated connections and interdependencies among different elements of complex systems, a new concept called nexus is proposed [11]. In recent years, many scholars have applied the nexus approach to conduct qualitative or quantitative research on the intricate connections of complex systems. The research methods of nexus mainly include input-output analysis [12,13], life cycle assessment [14,15], footprint [16,17], system dynamics model [18,19], network model [20,21], coupling coordination model [22,23], etc. These methods can conceptually explain and describe the status quo of nexus in terms of hidden connections, changing trends, causality, and coupling coordination levels.

Although the above research methods take into account the interconnectedness of complex systems, the human process is often biased as external factors to carry out scenario analysis of the co-evolution of complex systems [24]. Due to the high complexity of human activities, its impact on water resources and the environment is multi-level, multi-directional, and time-varying, which dominates changes in the ecological environment and resource endowments [8]. These changes occur in natural systems and are essentially caused by human value shifts on resource utilization and pollution control [25]. Therefore, it is necessary to make a more reasonable representation of the human process if the coevolutionary process of the human-water coupling system is to be better understood. However, less attention is paid to the dynamic mutual-feedback relationship among elements of the complex system, especially the impact of shifts in human values in this mutual-feedback relationship. In order to understand the dynamic characteristics and co-evolution of the human-water coupling system, a new theory, socio-hydrology, is proposed [26], which is intended to understand and portray the interaction of the human-water coupling system and its internal dynamics from social factors such as human consciousness, habits, and concepts. In recent years, research on the mutual feedback mechanism of the human-water coupling system has increasingly become an international frontier and hot issue, and the international academic community has carried out many studies about various complex systems guided by the socio-hydrology theory. For instance, there was fierce competition for water resources between regional economic development and ecological environment protection in the Tarim River basin in northwest China. This fierce competition was described as the phenomenon known as "pendulum swing", which is difficult for traditional hydrological models to capture [25]. Therefore, detailed modeling analyses of this phenomenon were conducted from the perspective of socio-hydrology [27]. In order to understand the co-evolution of human-flood systems, a stylized model was developed to portray the association among flood risk management strategies, social inequality and economic development [28]. With the dynamic coevolutionary perspective initiated by socio-hydrology serving as methodology, a system-dynamics model was constructed to understand and predict the coevolutionary behavior of the human-water coupling system dominated by water supply, power generation, ecological environment, and environmental awareness [29]. In essence, the variations of state variables are formulated by a series of coupled differential equations in such models, in which researchers can explore the mutual interactions and continuous feedbacks between humans and water, thereby providing insight into dynamics involved in the human-water coupling system [30].

In this study, inspired by the concept of socio-hydrology and the core idea of nexus, we aim to construct a co-evolution model to understand the coevolutionary behavior of the SWE coupling system. The remaining sections are organized as follows: The study area, the Yangtze River Economic Belt, as well as the data sources, are introduced in detail in Section 2. The SWE coupling system in the study area is first conceptualized, including the governing and response equations, and the coupling coordination degree is introduced in Section 3. In Section 4, the coevolutionary trajectories of the SWE coupling

system are analyzed and appraised. Furthermore, the model is calibrated and validated to investigate its reliability [31]. Finally, conclusions are drawn in Section 5. In summary, the objectives of this research are as follows: (1) Environmental sensitivity and environmental protection awareness are introduced to describe the impact of environmental changes on social values and their responses, in order to improve the cognition of the dynamic mutual feedback mechanism of the human-water coupling system. (2) Coupling coordination degree (called CCD in this research) is introduced to evaluate the coevolutionary similarities and differences of the SWE coupling systems in different study areas.

## 2. Study Area and Data Used

### 2.1. Study Area

The Yangtze River Economic Belt (called YREB in this research) is located in the hinterland of China, covering an area of about 2.05 million square kilometers (Figure 1), including 11 provincial administrative regions (Shanghai, Jiangsu, Zhejiang, Anhui, Jiangxi, Hubei, Hunan, Chongqing, Sichuan, Yunnan, and Guizhou), with more than 600 million people living in the area. After years of rapid development, the YREB has now become one of the new economic strategic supports and growth poles in China's regional development pattern. In 2020, the GDP of the YREB was about 47.2 trillion yuan, accounting for 46.4% of the country's GDP. However, despite achieving huge economic construction benefits, there are problems such as the continuous decline of environmental quality and the intensification of water resource constraints in the YREB. The causes of water-related problems in the YREB are complex, and there is a prominent contradiction between protection, restoration, governance, and development, with regional development sustainability and coordination seriously threatened. It is urgent for all regions in the YREB to coordinate the upgrading of industrial structure, ecological environmental protection, and efficient utilization of water resources with the overall high-quality development. Since the development of the YREB was elevated to a national strategy in 2014, how to coordinate social development, water resources utilization, and environmental protection, and achieve a holistic and systematic development of the YREB, has become a top priority. Therefore, scientific analysis of the coevolutionary relationship between water resource utilization, environmental protection, and social development has important practical significance for promoting the sustainable development and the harmonious symbiosis of human and water of the YREB.

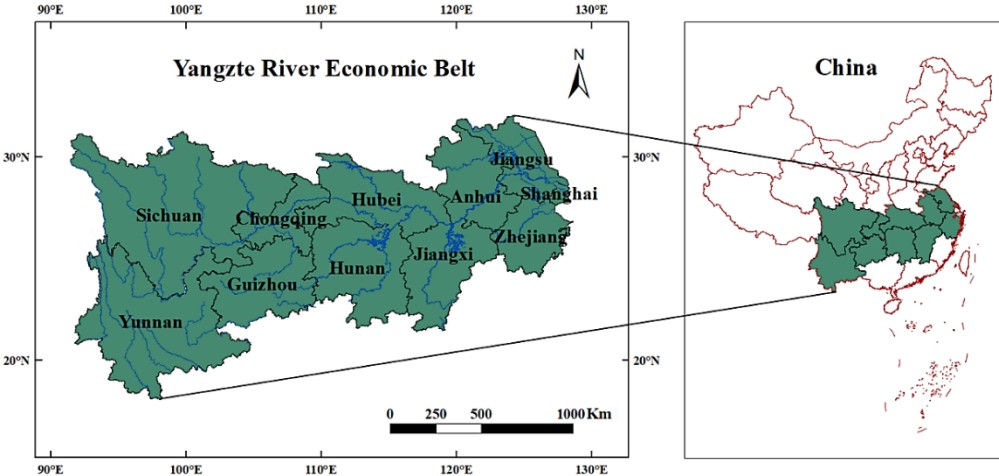

**Figure 1.** Location of the Yangtze River Economic Belt.

### 2.2. Data Used

Representative and universal data on water resources, society, and environment from 2006 to 2019 in the 11 provincial administrative regions of the YREB were collected for model calibration and testing. Socioeconomic data such as population, gross national product, and industrial added value are from the statistical yearbooks of the provincial

administrative regions in the YREB. Water resources data such as industrial water, domestic water, agricultural water, and ecological water are from the China Water Resources Bulletin. Environmental data such as total wastewater discharge, ammonia nitrogen emissions, and COD emissions are from the China Environmental Ecology Yearbook.

## 3. Model Description

### 3.1. Conceptual Model

The human process, the biogeochemical process of the water environment, and the water resource-utilization process are integrated into a whole from the perspective of socio-hydrology, which provides a new way to identify the causes and evolution of complex water problems. On this basis, a co-evolution model is constructed to simulate the social development process, human water-intake process and water-ecological environment process. Each provincial administrative region in the YREB is regarded as an SWE coupling system formed by the interconnection and interaction of society, water resources, and environment subsystems. It is assumed that an SWE coupling system is a water balance area, from which water resources can be extracted to meet the needs of industry, livelihood, agriculture, ecology, etc. During the process of water withdrawal, pollutants enter the environment with the water cycle. Social factors such as adjustments in economical structural, changes in people size, fluctuations in human consciousness, and progress in science and technology will have a dynamic interaction with water resources utilization and ecological environment evolution.

Based on the above concept, the corresponding state variables are determined to describe the internal process of the SWE coupling system. The mutual feedback relationship between the elements in the SWE coupling system is shown in Figure 2. In the model, $U_i$ is used to represent the water consumption of the $i$-th use in the water resources subsystem; $S$ is used to represent the development status of social indicators in the society subsystem; $C_j$ is used to represent the discharge of the $j$-th pollutant in the environment subsystem. In addition, three response variables are used to assist in establishing the constitutive relationship of state variables among three subsystems: environmental sensitivity ($V$), environmental protection awareness ($E$), and technology level ($T$). The constitutive relationship between state variables is related by response variables in positive or negative feedback. The coevolutionary process of the SWE coupling system is quantified into the following two aspects: the governing equations of each state variable, and the response equations that characterize the driving or hindering effects of different variables. Simultaneously, the coupling coordination degree (CCD) is used to measure the coevolutionary level of the SWE coupling system.

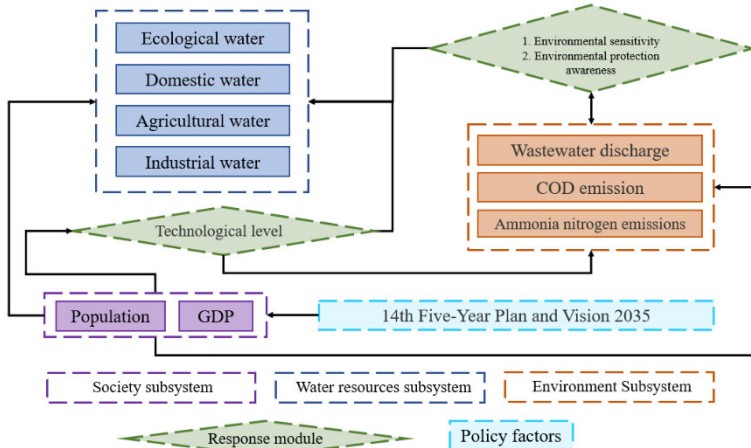

**Figure 2.** The mutual feedback relationship within the coevolution model. The variables of the society subsystem, water resources subsystem, environment subsystem, response module and policy factor are shown in purple, blue, orange, green and cyan, respectively.

*3.2. Governing Equations*

3.2.1. Society Equations

The "14th Five-Year Plan for National Economic and Social Development of the People's Republic of China and Outline of Vision 2035" (hereinafter referred to as the "Plan and Outline") set the tone for China's economic and social development in the next five to fifteen years. Therefore, the economic and social state variables $S$ in the simulated area are predicted according to the "Plan and Outline" and the corresponding plans of each provincial administrative region:

$$\frac{dS}{dt} = f(S, \delta, t) \tag{1}$$

where $S$ is the economic and social vector that evolves with time $t$, including GDP, population, industrial added value, etc. The parameter vector $\delta$ is the policy factor, which is selected with reference to the "Plan and Outline" and the corresponding plans of each provincial administrative region. $f(\cdot)$ is a matrix-valued function based on the system dynamics theory. $S, \delta$ and $f(\cdot)$ are all multi-dimensional, $S = (S_1, S_2, \ldots, S_n)^T \in R^n$, $\delta \in R^m$, $R^n \times R^m \to R^n$, where $n$ and $m$ are the number of economic and social variables and policy factors, respectively.

3.2.2. Water Resources Equations

The total water consumption $U$ in the forecast area can be expressed as:

$$U = \sum U_i \tag{2}$$

where $U_i$ is the water consumption for living, industry, agriculture, and ecology, which is expressed by the following differential equation:

$$\frac{dU_i}{dt} = (\gamma_i + X_i)U_i \tag{3}$$

where $\gamma_i$ is the endogenous growth rate of the $i$-th water consumption, and $X_i$ is the response equation of the state variable $U_i$, which is introduced in the Section 3.3, the same below.

3.2.3. Environmental Equations

The total amount of wastewater discharge, chemical oxygen demand (COD) emission, and ammonia nitrogen emission are used as the state variables of the environmental subsystem. The discharge amount $C_j$ of the $j$-th pollutant is expressed by the following differential equation:

$$\frac{dC_j}{dt} = (\sigma_j + X_j)C_j \tag{4}$$

where $\sigma_j$ is the endogenous growth rate of the $j$-th pollutant emission, and $X_j$ is the response equation of the state variable $C_j$.

*3.3. Response Equations*

With the development of economy and society, changes in scientific and technological level and human consciousness have caused a far-reaching and extensive impact on water consumption and ecological environment quality, endowing social connotations in the natural attributes of water resources and ecological environment. In order to describe these impacts more intuitively, these impacts are decomposed into several components, each of which represents the contribution of the corresponding impact factor, so as to obtain the response equation of the specified state variable. The response equation $X_k$ for the $k$-th state variable is expressed as:

$$X_k = \sum_{l=1}^{L} \mu(x_l) \cdot w_l(x_l) \cdot G_l(x_l) \tag{5}$$

In the formula, $L$ is the number of impact factors, where $\mu(x_l)$ is an indicator function, indicating whether the impact factor $x_l$ has a positive effect or a negative effect:

$$\mu(x_l) = \begin{cases} 1 & \text{positive effect} \\ -1 & \text{negative effect} \end{cases} \tag{6}$$

where $w_l(x_l)$ is the weight of the influence factor $x_l$ in the response equation, $\sum w_l(x_l) = 1$. $G_l(x_l)$ is a normalized function that controls the impact factor $x_l$ in [0,1].

Each impact factor is described in detail as follows:

### 3.3.1. Environmental Sensitivity and Environmental Protection Awareness

In the socio-hydrology conception, the co-evolution of the human-water coupling system is affected by human activities, and the preference of human activities depends on the change of community sensitivity, which is the main driving factor for the co-evolution of the human-water coupling system [10]. The more a community feels that its quality of life is threatened, the more likely it is to show a high degree of sensitivity to small changes in factors that may subsequently negatively affect its quality of life. Conversely, the less a community perceives its quality of life to be threatened, the less likely it is to be sensitive (and therefore responsive) to small changes in these variables. In many studies, the community sensitivity is simulated as internal variables of the coupling system to capture human society's perception level of environmental changes and serve as the basis for driving the evolution of the human-water coupling system [32,33]. However, the impact of changes in community sensitivity on the direction of system evolution is generally not considered in the traditional system dynamics method, and there is a lack of expressions for the mutual feedback mechanism between the natural system and the human system. In addition, because the input variables in the model are only based on the social development indicators in the "Plan and Outline", there are no quantitative indicators in the relevant policies of water resources management and environment governance as input variables in the medium- and long-term. Therefore, environmental sensitivity and environmental protection awareness are introduced as the driving factors for changes in water consumption and pollutant emissions in this model, and are used to simulate the changes in community sensitivity, portraying the dynamic feedback in the SWE coupling system. It is assumed that the environmental sensitivity and environmental protection awareness are closely related to the discharge of water pollutants in this model. The equations [34] describing environmental sensitivity $V$ and environmental protection awareness $E$ are as follows:

$$\frac{dV}{dt} = \left( \sum_{j=1}^{K} w_{C_j} \cdot \widetilde{C}_j \right) V \tag{7}$$

$$V^* = \frac{\frac{dV}{dt}}{V_{max} - \overline{V}} \tag{8}$$

$$E = \begin{cases} \frac{V^*}{1+V^*} & V^* \geq V_{crit} \\ 0 & V^* < V_{crit} \end{cases} \tag{9}$$

where $\widetilde{C}_j = \Delta\overline{C_j}/\overline{C_j}$ is the relative changes in $j$-th pollutant emission, $\overline{C_j}$ is the three-year moving average, and $\Delta\overline{C_j}$ are calculated as the differences between current three-year average values and previous 3-year average values. These averaging windows are used to reflect the time-lag effect between changes in water pollutants and environmental sensitivity [34]. $w_{C_j}$ is a weighting factor to quantify the relative importance of different pollutant emissions to the rate of change in environmental sensitivity, $\sum w_{C_j} = 1$. $V^*$ is a normalized environmental sensitivity, which is used in consideration of the effect of the baseline sensitivity on the timing and magnitude of response behavior. For instance, the higher the environmental sensitivity level is, the more direct and severe its response behavior would be when the environmental sensitivity increases. $V_{max}$ is an assumed

maximum environmental sensitivity and we set its value to 200 with reference to previous research [35]. $\overline{V}$ is the three-year moving average of environmental sensitivity, which is used to quantify a period of the social memory. The term $V_{max} - \overline{V}$ is used only to quantify an incremental change in environmental sensitivity as the baseline sensitivity increases. In addition, normalized environmental sensitivity $V^*$ is proposed to translate to environmental protection awareness $E$ following a sigmoidal function [25], as shown in Equation (9). $V_{crit}$ is a critical environmental sensitivity, only above which can the environmental sensitivity be translated into environmental protection awareness and corresponding measures be stimulated to take enviro-centric action.

### 3.3.2. Technology Level

The technology level reflects the ability of a community to develop and utilize water resources and manage the ecological environment. The improvement of the technology level can play a positive role in improving water-use efficiency and the water-pollution control process. Therefore, the technology level $T$ is used to quantify its impact on water resources and the environment, and its equation [30] is as follows:

$$T = \theta \overline{W} \tag{10}$$

where $\theta$ is the technology conversion coefficient, and $\overline{W}$ is the 3-year moving average of the per capita GDP of the simulated region, which is used to characterize the continuous impact of socioeconomic development on the technology level [30].

### 3.4. Coupling Coordination Degree

The coupling coordination degree (CCD) is used to quantify the microscopic synergy and macroscopic order within nonlinear complex systems, and has been widely used to characterize the spatiotemporal differentiation characteristics and coupling interaction of coupling systems [36]. The size of the CCD can indicate whether each part in the coupling system promotes or restricts each other [37]. The greater the CCD is, the higher the level of coordinated development of the coupling system is. Following the principles of dynamics, science and objectivity, the CCD evaluation indicator system of the SWE coupling system is shown in the Table 1.

**Table 1.** The CCD evaluation indicator system of the SWE coupling system.

| Subsystem | Indicators | Property |
|---|---|---|
| Society | GDP per capita | Positive |
| | Population density | Negative |
| | environmental sensitivity | Negative |
| Water resources | Water consumption per 10,000 yuan of GDP | Negative |
| | Water consumption per 10,000 yuan of industrial added value | Negative |
| | Agricultural water consumption per unit area | Negative |
| | Domestic water consumption per capita | Negative |
| | proportion of ecological water consumption | Positive |
| Environment | Wastewater discharge intensity | Negative |
| | COD emission intensity | Negative |
| | Ammonia nitrogen emission intensity | Negative |

Note: If the indicator type is positive, the larger the indicator value is, the better the situation is. If the indicator type is negative, the smaller the indicator is, the better the situation is.

Referring to the calculation method of CCD proposed by existing research results [36,38], the contribution of the $j$-th variable $\alpha_{ij}$ of the $i$-th subsystem to the CCD of the SWE coupling system is calculated by the following formula:

$$\xi_{ij} = \begin{cases} \frac{\alpha_{ij} - \alpha_{imin}}{\alpha_{imax} - \alpha_{imin}} & \alpha_{ij} \leq \overline{\alpha_{ij}} \\ \frac{\alpha_{imax} - \alpha_{ij}}{\alpha_{imax} - \alpha_{imin}} & \alpha_{ij} > \overline{\alpha_{ij}} \end{cases} \tag{11}$$

where $\xi_{ij} \in (0,1)$, $\alpha_{imax}$ and $\alpha_{imin}$ are 1.01 times the upper limit and 0.99 times the lower limit of the $i$-th subsystem (to avoid the limit cases of 0 and 1 in the value of $\xi_{ij}$), and $\overline{\alpha_{ij}} = \frac{\sum_{j=1}^{J} \alpha_{ij}}{J}$, $J$ is the total number of variables in $i$-th subsystem.

The cooperative degree $H_i$ of the $i$-th subsystem is calculated by the following formula:

$$H_i = \sum_{j=1}^{J} w_j \xi_{ij} \quad (i = 1, 2, 3) \tag{12}$$

where $w_j$ is the weight of the $j$-th variable of the $i$-th subsystem, which is calculated by the entropy weight method. As an objective weighting method, the entropy weight method avoids the deviation caused by human factors, and has been widely used in the fields of engineering science [39]. The larger $H_i$ is, the higher the coordination degree of the $i$-th subsystem is.

The *CCD* in SWE coupling system is calculated by the following formula:

$$CCD = \sqrt{CQ}, \quad Q = \sum_{i=1}^{n} w_i H_i, \quad C = n \left[ \frac{\prod_{i=1}^{n} H}{\left(\sum_{i=1}^{n} H\right)^n} \right]^{1/n} \tag{13}$$

where $C$ is the coupling degree among the subsystems, $Q$ is the overall comprehensive evaluation index of the SWE coupling system, and $w_i$ is the weight of the $i$-th subsystem. Assuming that the three subsystems are equally important, the weight of each subsystem is 1/3. $n$ is the number of subsystems, taken as 3.

## 4. Results and Discussion

### 4.1. Model Validation

The research object of this model is each provincial administrative region in the YREB. The model runs from 2006 to 2035. The model parameters are calibrated using the SCE-UA algorithm, which is a compound optimization algorithm often used for global parameter calibration [40]. Due to the relatively short historical observation data series, the data set from 2006 to 2019 was adopted to validate and calibrate the model. The period from 2020 to 2035 is the prediction stage of the model, in which policy factors affecting the changes in economic and social indicators are selected with reference to "Plan and Outline", as well as the corresponding planning of each provincial administrative region. Due to the large number of variables in the model, some representative key variables (GDP, population, total water consumption and total wastewater discharge) were selected to verify the model. As shown in Figures 3 and 4, the fitness of the simulation results and statistical data on the trend is acceptable. During the verification period, the relative error average and median of the representative indicators are generally within 10%, which could indicate that the model has good system stability and simulation effect. Additionally, the sensitivity degree to changes in parameters was analyzed with reference to the sensitivity calculation methods [41]. Four important parameters were selected for sensitivity analysis, namely the technology conversion coefficient ($\theta$), the GDP growth rate ($\delta_1$), the population growth rate ($\delta_2$) and industrial added-value growth rate ($\delta_3$). Total water consumption ($U$) and total wastewater discharge ($C_1$) were selected as the target variables for the sensitivity analysis. The results of the sensitivity analysis are shown in Figure 5, and it can be seen that most of the sensitivities are less than 10% and only a small number of the target variables are more sensitive to parameter adjustments, which indicates that the model is insensitive to parameter changes and relatively robust. That is to say, it can effectively simulate the real process of the system.

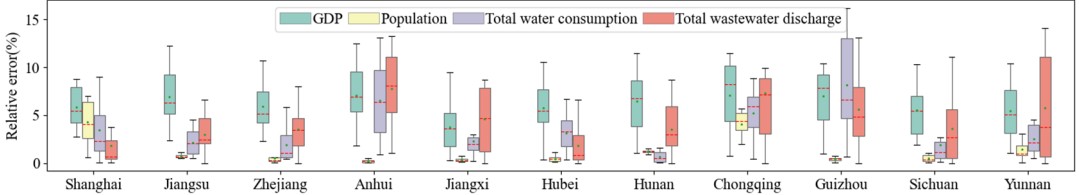

**Figure 3.** Box diagram of relative error.

**Figure 4.** Simulation results of co-evolution of water resources-society-environment in 11 provincial administrative regions of the YREB. Note: In order to reflect the change in the proportion of water use during the forecast period, the simulated values of various water consumption results after 2020 are represented with corresponding colors with lower saturation.

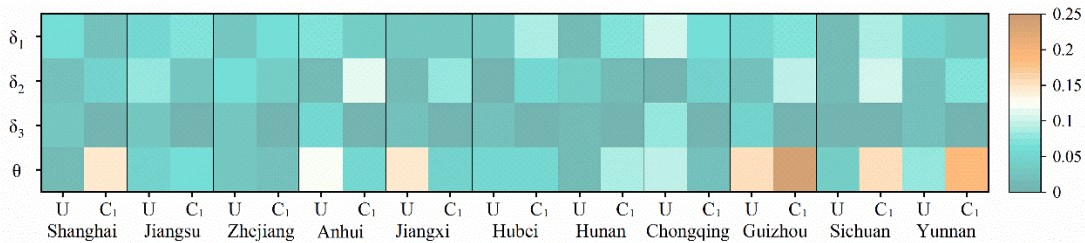

**Figure 5.** Sensitivity analysis results.

### 4.2. Co-Evolution of the SWE Coupling System

A variety of factors produce complex interactions, which are transmitted through environmental sensitivity, environmental protection awareness, and technological level, and accumulate over time under the framework of co-evolution. Therefore, the SWE coupling system of each provincial administrative region presents different co-evolution paths. Taking the simulation results of the total wastewater discharge in Sichuan and Hubei in Figure 4 as an example, the total wastewater discharge in Sichuan Province gradually increased from 2006 to 2015, and the environment health status showed a deteriorating trend. This dynamic response process is shown in the model as follows: the environmental sensitivity gradually increased and exceeded the threshold in 2015, thus awakening environmental protection awareness, and corresponding water pollution control measures were stimulated, resulting in a brief decline in total wastewater discharge from 2016 to 2019. Although control measures have slowed the growth of total wastewater discharge, it is not enough to eliminate the impact of increased wastewater discharge caused by the expansion of water use. The total wastewater discharge will continue to rise during the forecast period and reach its peak in 2025, and the environmental sensitivity will once again exceed the threshold, and a stronger environmental protection awareness will be stimulated. Driven by both stronger control measures and more advanced water pollution control technologies, the total wastewater discharge in Sichuan Province will drop significantly from 2025 to 2027. During the simulation and forecast period, the evolution of the total wastewater discharge in Hubei Province first showed an increasing trend, and reached its peak in 2015, when the environmental sensitivity reached a high level and the environmental protection awareness was stimulated. With the effectiveness of the corresponding control measures, the total wastewater discharge in Hubei Province has gradually decreased. The difference in the evolution trajectories of the total wastewater discharge between the two provinces is the manifestation of the nonlinear response relationship between the variables of the SWE coupling system, which reflects the variability and complexity of the macroscopic space-time behavior of the coupling system.

Although the co-evolution trajectories of each provincial administrative region of the YREB are different, there are certain commonalities in the direction of evolution under the constraints of policies, regulations and laws. For instance, ammonia nitrogen emissions and COD emissions in all provincial administrative regions will show a downward trend in the simulation forecast period. Combined with the analysis of the evolution trajectory of environmental sensitivity in Figure 4, the evolution trajectory of environmental sensitivity in each provincial administrative region is generally in the direction of "rising-fluctuation-decline" during the simulation and forecast period. The dynamic fluctuation process of environmental sensitivity is the embodiment of the mutual feedback between the society subsystem and the environment subsystem. When environmental pollution in the study area threatens the quality of life (environmental sensitivity exceeds a threshold), environmental protection awareness is triggered and human society takes corresponding measures to improve the living environment conditions. For example, each time environmental protection awareness is aroused, the total wastewater discharge fluctuates downwards accordingly, which is the response of human society in the face of the threat of environmental degradation. The complexity of the social development of each provincial administrative region has resulted in differences in the fluctuation range and peak time of

their environmental sensitivity and the frequency, intensity and duration of environmental protection awareness. However, with the improvement of pollution control technology and the guidance of relevant policies, the governance and regulation of the environment has become more legitimate and effective. The maximum environmental sensitivity of each provincial administrative region basically appeared between 2006 and 2025, and will have dropped below 100 in 2035. The consistency in the evolution direction of environmental sensitivity reflects the changing process of the environment of each provincial administrative region from good to bad, and then from bad to good. At the end of the forecast period, the environmental sensitivity decreased, which not only reflected that communities are less sensitive to environmental changes in a good environment, but also reflects the positive trend of environmental changes in each provincial administrative region of the YREB at the macro level.

Since the launch of the development strategy of the YREB, 11 provincial administrative regions have accelerated the promotion of eco-friendly development, with industrial structure and water resources utilization continuously optimized. As shown in Figure 6, the water consumption per 10,000 yuan of GDP and the water consumption per 10,000 yuan of industrial added value in each provincial administrative region varies greatly in 2010. With the advancement of the green development pattern, the water consumption per 10,000 yuan of GDP and the water consumption per 10,000 yuan of industrial added value in each provincial administrative region will gradually decrease during the forecast period, and the gap between provincial administrative regions will gradually shrink. By 2035, the water consumption per 10,000 yuan of GDP and the water consumption per 10,000 yuan of industrial added value in each provincial administrative region will have converged to less than 100 m$^3$, which reflects a good expectation of continuous improvement of regional water use efficiency under the guidance of the technological level improvement and the policy factors related to eco-friendly development.

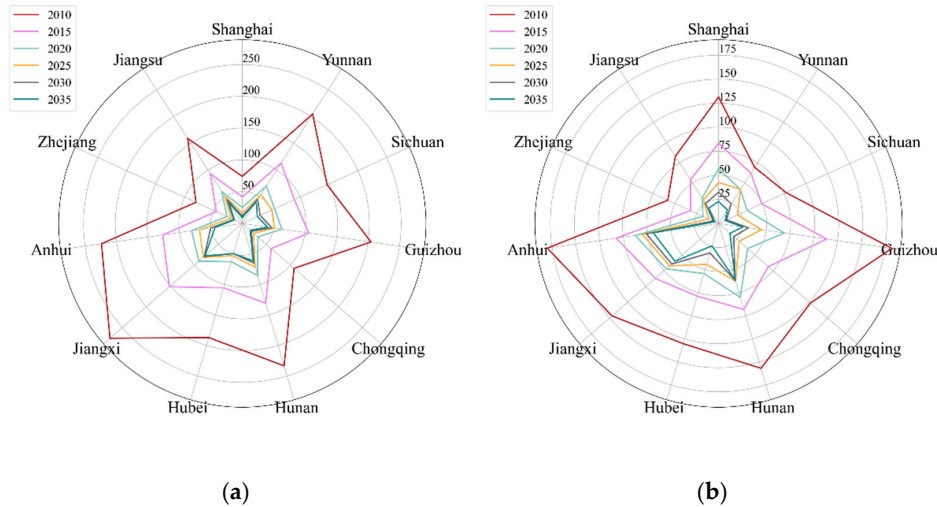

(a)                                                                                (b)

**Figure 6.** Water use indicators of 11 provincial administrative regions of the YREB. (**a**) Water consumption per 10,000 yuan of GDP (m$^3$). (**b**) Water consumption per 10,000 yuan of industrial added value (m$^3$).

The calculation results of the CCD of the 11 provincial administrative regions in the YREB are shown in Figure 7. The CCD trajectory of the SWE coupling system in each provincial administrative region is relatively similar, basically showing the trend of "stable-up-stable". From 2006 to 2014, the demand for water resources in various provincial administrative regions gradually increased, and the negative impact of social development on the environment became increasingly apparent. The degree of coordinated development among the three subsystems of society, water resources, and environment is relatively low, and the CCD of each provincial administrative region grows slowly, basically in the range of

0.2 to 0.5. After the YREB was established as a major national development strategy in 2014, the provincial administrative regions of the YREB have comprehensively strengthened the rational development and effective utilization of water resources, and intensified efforts to manage the ecological environment. The growth rate of the CCD of each provincial administrative region has gradually accelerated, transitioning from slow to rapid growth. During the 14th Five-Year Plan period, the CCD of each provincial administrative region is expected to continue to grow, and the comprehensive social and economic development, water resources utilization and environment protection show a trend of coherent and substantial improvement. It is expected that the CCD of each provincial administrative region will reach a high level around 2025, reflecting that the 14th Five-Year Plan has played a good policy-supporting role in the coordinated development of the YREB.

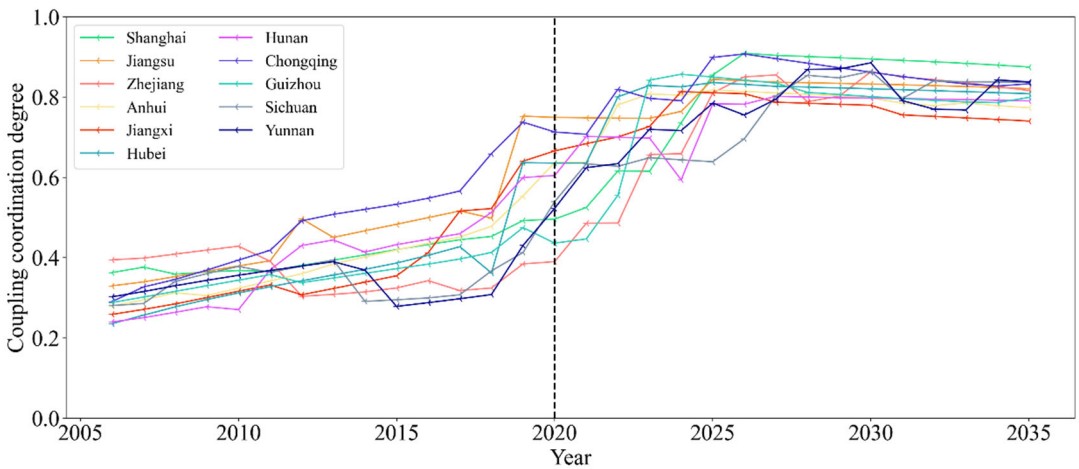

**Figure 7.** The trend of CCD of 11 provincial administrative regions of the YREB.

## 5. Conclusions

The development of the YREB is a major development strategy measure combining China's regional coordinated development and opening up to the outside world in the new era. The coevolution of its social development, water resources utilization, and environmental protection is of great significance for China to move towards a medium-to-high level of development. Based on this background, a coevolution model is constructed to explore the coevolutionary process of the SWE coupling system, which fully considers the influence of social factors such as environmental sensitivity, environmental protection awareness and technological level on the co-evolution process. The provincial administrative regions in the YREB are used as analysis units in the co-evolution model, and simulations and predictions are made for their social development process, water resource utilization process, and water pollution discharge process. The conclusions are as follows: The prediction results of the SWE coupling system have high accuracy, which can better reveal the coupling and mutual-feedback mechanism under the influence of multiple factors in the co-evolution. Although the variables in the SWE coupling system of the provincial administrative regions of the YREB show different coevolution trajectories during the forecast period, the water-use efficiency and the environment quality of each provincial administrative region of the YREB have gradually improved with the awakening of environmental protection awareness and the improvement of water conservation and pollution control ability under the policy support of the "Plan and Outline". During the 14th Five-Year Plan period, the CCD of each provincial administrative region of the YREB has increased significantly, reflecting that the 14th Five-Year Plan has played a good policy supporting role in the coordinated development of the YREB. The environmental sensitivity can be reduced to a lower level around 2035 after the fluctuation, reflecting the positive trend of environmental changes of the YREB. The environmental sensitivity and environmental protection awareness play a key role in simulating the co-evolutionary trajectory of the SWE coupling system, which can

provide a reference for the follow-up research on the dynamic mutual feedback mechanism of the human-water coupling system.

Although the model constructed by this research tries to fully couple variables in society, water resources, and environment subsystems of the provincial administrative regions of the YREB, it is still only a simple description of a huge and complex coupling system and a trend prediction for the coordinated development of the provincial administrative regions of the YREB in the context of macro policies. Due to the complexity of the coevolution of the coupling system, the precise expression of the co-evolution law needs to be further studied.

**Author Contributions:** Conceptualization, H.L. and X.Z.; Data curation, K.W.; Formal analysis, S.T.; Funding acquisition, J.X.; Investigation, S.T.; Methodology, H.L.; Project administration, J.X.; Supervision, X.Z.; Validation, X.X.; Visualization, H.L.; Writing—review and editing, X.X. All authors have read and agreed to the published version of the manuscript.

**Funding:** This research is supported by the National Key Research and Development Program of China (NO.2019YFC0408901) and the National Natural Science Foundation of China (No. 41890823).

**Institutional Review Board Statement:** Not applicable.

**Informed Consent Statement:** Not applicable.

**Data Availability Statement:** The data that support the findings of this study are available from the authors upon reasonable request.

**Conflicts of Interest:** The authors declare no competing interests.

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
