# Peer review of "A Coevolution Model of the Coupled Society—Water Resources—Environment Systems: An Application in a Case Study in the Yangtze River Economic Belt, China"

_water, doi:10.3390/w14152449_

Round 1

Reviewer 1 Report

The paper needs major revisions:

1-What is the "SWE coupling system" in the introduction section?

2-What are the major challenging issues of SWE coupling system in hydrological modeling systems?

3-In order to improve the results section, authors need to compute uncertainty, reliability and sensitivity analyses for the SWE coupling system.

4-Introduction literature needs adding literature review:A comprehensive uncertainty analysis of model-estimated longitudinal and lateral dispersion coefficients in open channels

Reviewer 2 Report

The study is very interesting and well describes, this represent a new approach to the problem of the management of water resources trying to take in consideration many different aspect.

Reviewer 3 Report

1. The equations used by author should be properly referenced.

2. Which method has been used for estimation of impact factors. It should be mentioned by authors in the manuscript.

3. [Line 218] How authors decided quality of life in the study area was threatened. Did any survey been done. Authors should mention in the manuscript.

4. Interaction of water and society and its response depend on the changes in technological interventions also. It depends on the awareness of people also. Do these parameters has been taken into account in this manuscript.

Round 2

Reviewer 1 Report

Accept as is

Author Response

Once again, thank you very much for your constructive comments on the improvement of the manuscript.